

# Searching for a signature involving 10 genes to predict the survival of patients with acute myelocytic leukemia through a combined multi-omics analysis

Haifeng Zhuang[1,*], Yu Chen[2,*], Xianfu Sheng[1], Lili Hong[1], Ruilan Gao[1] and Xiaofen Z

[1] The First Affiliated Hospital of Zhejiang Chinese Medical University, Hang Zhou, China
[2] Hangzhou Medical College, Hang Zhou, China
[3] Hangzhou Fuyang Hospital of Traditional Chinese Medicine, Hang Zhou, China
[*] These authors contributed equally to this work.

Corresponding author
Haifeng Zhuang,
zhuanghaifeng5@126.com

## ABSTRACT

**Background**. Currently, acute myelocytic leukemia (AML) still has a poor prognosis. As a result, gene markers for predicting AML prognosis must be identified through systemic analysis of multi-omics data.

**Methods**. First of all, the copy number variation (CNV), mutation, RNA-Seq, and single nucleotide polymorphism (SNP) data, as well as those clinical follow-up data, were obtained based on The Cancer Genome Atlas (TCGA) database. Thereafter, all samples ($n = 229$) were randomized as test set and training set, respectively. Of them, the training set was used to screen for genes related to prognosis, and genes with mutation, SNP or CNV. Then, shrinkage estimate was used for feature selection of all the as-screened genes, to select those stable biomarkers. Eventually, a prognosis model related to those genes was established, and validated within the GEO verification ($n = 124$ and 72) and test set ($n = 127$). Moreover, it was compared with the AML prognosis prediction model reported in literature.

**Results**. Altogether 832 genes related to prognosis, 23 related to copy amplification, 774 associated with copy deletion, and 189 with significant genomic variations were acquired in this study. Later, genes with genomic variations and those related to prognosis were integrated to obtain 38 candidate genes; eventually, a shrinkage estimate was adopted to obtain 10 feature genes (including FAT2, CAMK2A, TCERG1, GDF9, PTGIS, DOC2B, DNTTIP1, PREX1, CRISPLD1 and C22orf42). Further, a signature was established using these 10 genes based on Cox regression analysis, and it served as an independent factor to predict AML prognosis. More importantly, it was able to stratify those external verification, test and training set samples with regard to the risk ($P < 0.01$). Compared with the prognosis prediction model reported in literature, the model established in this study was advantageous in terms of the prediction performance.

**Conclusion**. The signature based on 10 genes had been established in this study, which is promising to be used to be a new marker for predicting AML prognosis.

## INTRODUCTION

Acute myeloid leukemia (AML) is an aggressive hematological neoplasm commonly seen in adulthood. It is characterized by immature myeloid hematopoietic cell accumulation, particularly within the bone marrow (*Kouchkovsky & Abdul-Hay, 2016*). Peripheral blood is typically affected, which can result in the infiltration of the malignancy to the liver, spleen, lymph nodes, central nervous system (CNS), and skin. AML is commonly treated using post-remission treatment and intensive induction chemotherapy, However, patient survival has not improved substantially over the past thirty years (*Lai, Doucette & Norsworthy, 2019*). Remarkable remission rates can be initially attained through chemotherapy in most AML patients but it is rare to see a complete elimination of the disease (*Drummond, 2019*). Alternative treatments like the CD33 chimeric antigen receptor (CAR)-based T-cell treatment that targets CD33, combined with allogeneic hematopoietic cell transplantation (HSCT) (*Kenderian et al., 2015*; *Walter, 2018*; *Bhatt, 2019*) may also be used to treat AML. However, there is still a risk of recurrence in 75% of cases 5 years after the initial diagnosis. It is necessary to find a biomarker that can be used to predict the prognosis of AML to assist clinicians and be used for individualized healthcare.

The current prognosis criteria for AML is highly dependent on age, cytogenetic abnormality, white blood cell (WBC) count at the time of diagnosis, and the identification of the molecular genetic disorder specific to AML (*Papaemmanuil et al., 2016*). Many studies have been conducted to examine the genomic landscape of AML and to improve our understanding of its development. However, it is difficult to apply these findings to the clinical treatment of AML and although great strides have been made in stratifying risk, patients that exhibit fewer risk factors may relapse (*Rollig et al., 2011*). Biomarkers are divided into multiple types, such as single molecules used as indicators to independently predict prognosis (PDLIM (*Cui et al., 2019*), PDE7B (*Cao et al., 2019*), and DOCK2 (*Hu et al., 2019*)), and gene markers established through different prognostic genes using high throughput analysis of gene expression patterns. *Huang, Liao & Li (2017)* proposed a risk score system based on an 11-gene signature to predict and assess the prognosis of AML. *Tang et al. (2019)* used the least absolute shrinkage and selection operator Cox regression to identify the signature based on 10 lncRNAs within lncRNA expression data. These studies tested the contemporary gene signatures with an independent external data set but their findings revealed low values of area under the curve (AUC) at 3–7 years. The identification of the gene signature or biomarker predicting AML prognosis remains challenging and the results should be verified in more cohorts. We must identify the gene signals related to the prognosis for AML using bioinformatic analyses of specific biological activities.

We used a systemic multiomics strategy to effectively identify a creditable gene signature related to AML prognosis and search for genetic markers related to AML. We acquired the data for all AML cases, the copy number variation (CNV), single nucleotide polymorphism (SNP), gene mutation, and gene expression profile data from the GEO and TCGA databases. A signature for identifying prognostic markers was established based on 10 genes by combining transcriptomic data with genomic data. The predictive capacity was validated using external verification and internal test sets, respectively. Compared with

other prognosis prediction models from the literature, our model was more accurate and advantageous in predicting the prognosis for patients with survival of over 5 years. According to GO analysis, the signature based on 10 genes took part in vital AML-related pathways and biological processes. Similar findings were obtained through gene set enrichment analysis (GSEA), suggesting that the as-constructed 10-gene signature could efficiently estimate the prognostic risk among AML cases and may improve the understanding of the AML prognostic mechanism at the molecular level.

## MATERIAL AND METHODS

### Data extraction and processing

The data used in this study are publicly accessible at The Cancer Genome Atlas (TCGA) (search terms: TCGA-LAML) and NCBI GEO (accession number: GSE37642, GSE12417) databases. We used the UCSC cancer browser to download CNV, clinical follow-up, and TCGA RNA-Seq data of the SNP 6.0 chip. That information can be found at: https://xenabrowser.net/datapages/?cohort=GDC%20TCGA%20Acute%20Myeloid%20Leukemia%20(LAML)&removeHub=https%3A%2F%2Fxena.treehouse.gi.ucsc.edu%3A443. The mutation annotation file (MAF) was collected based on the GDC client and GSE37642 (*Herold et al., 2018*) and GSE12417 (*Metzeler et al., 2008*) expression patterns. Clinical follow-up data were also obtained from the GEO database. 229 AML samples with sufficient follow-up data were screened for TCGA RNA-Seq data and were randomized as two groups, the test set ($n = 107$) and the training set ($n = 102$). The GSE37642 ($n = 124$) and GSE12417 ($n = 72$) data sets were used as external verification sets. Table 1 shows the sample details of every group. The mini format files were downloaded from the GEO platform to be used for GEO data processing and the probe ID was converted to the gene symbol according to the background file. We calculated the average value of multiple genes corresponding to a single probe, the probes corresponding to multiple genes were eliminated, and the expression spectrum matrix was further normalized.

### Univariate Cox proportional hazard regression analysis

Univariate Cox proportional hazard regression analysis was conducted to identify genes whose expression levels correlated with patient overall survival (OS) in the training set with $P < 0.01$.

### Analysis of CNV data

GISTIC was used to detect frequent and potentially overlapping recurrent events (*Pecina-Slaus et al., 2019*). Consequently, GISTIC 2.0 was adopted for use with CNV data from TCGA to determine the significantly deleted or amplified genes at $p < 0.05$ and fragments that had $>0.1$ deletion or amplification length.

### Analysis of gene mutation

Mutsig 2.0 (*Rahane, Kutzner & Heese, 2019*) software was used to recognize genes with significant mutations based on the MAF of TCGA mutation data at $P < 0.05$.

**Table 1  Clinical information statics of each groups of data sets.**

| Characteristic | | TCGA training datasets ($n = 102$) | TCGA datasets ($n = 127$) | GSE12417 ($n = 72$) | GSE37642 ($n = 124$) |
|---|---|---|---|---|---|
| Age(years) | ≤60 | 63 | 80 | 30 | 69 |
| | >60 | 39 | 47 | 42 | 55 |
| Survival Status | Living | 42 | 51 | 32 | 38 |
| | Dead | 60 | 76 | 40 | 86 |
| Gender | Female | 48 | 60 | / | / |
| | Male | 54 | 67 | / | / |
| FAB | M0 | 10 | 12 | 1 | 7 |
| | M1 | 25 | 29 | 20 | 27 |
| | M2 | 23 | 31 | 33 | 45 |
| | M3 | 12 | 13 | 0 | 7 |
| | M4 | 21 | 27 | 9 | 14 |
| | M5 | 9 | 12 | 5 | 15 |
| | M6 | 1 | 2 | 3 | 7 |
| | M7 | 1 | 1 | 0 | 1 |
| Bone marrow blast percentage | >80 | 41 | 48 | / | / |
| | ≤80 | 61 | 79 | / | / |
| White blood cell count | >20 | 45 | 55 | / | / |
| | ≤20 | 57 | 72 | / | / |

## Prognosis-related gene signature construction

The lasso cox regression was used to refine the identified prognostic genes using the glmnet function of R package (*Engebretsen & Bohlin, 2019*). The MASS function of R package was used to carry out stepwise regression analysis in accordance with the Akaike information criterion and obtain the eventual 10-gene risk model. The formula used was:

$$\text{RiskScore}_{10} = 3.1175 * \exp^{\text{CAMK2A}} - 1.3247 * \exp^{\text{FAT2}} - 1.7172 * \exp^{\text{GDF9}}$$
$$- 0.907 * \exp^{\text{TCERG1}} - 0.4339 * \exp^{\text{DOC2B}} + 0.4833 * \exp^{\text{PTGIS}}$$
$$+ 0.426 * \exp^{\text{PREX1}} + 1.4097 * \exp^{\text{DNTTIP1}} + 4.6867 * \exp^{\text{C22orf42}}$$
$$+ 0.8599 * \exp^{\text{CRISPLD1}}.$$

The risk score values of the $z$-score were normalized and samples with the processed $z$-score value of $>0$ were classified as the high-risk group, while those $<0$ were the low-risk group.

## Functional enrichment analyses

The clusterprofiler (v3.8.1) (*Yu et al., 2012*) was used to perform Kyoto Encyclopedia of Genes and Genomes (KEGG) and Gene Ontology (GO) pathway enrichment analyses on genes to recognize the enriched KEGG pathways and GO terms among the three categories: cellular component (CC), molecular function (MF), and biological processes (BP). The false discovery rate (FDR) value of $<0.05$ indicated statistical significance. The expression

matrix of genes between different samples was converted into the expression matrix of gene sets to evaluate which metabolic pathways were enriched. The correlation between the risk scores and pathways was further calculated using Pearson correlation analysis. Signaling pathways with a correlation coefficient of >0.35 were considered to be related to the risk score.

## Statistical methods

The median risk score of every data set was used as the threshold to plot the Kaplan–Meier (KM) curves and the survival risks were compared in a high-risk group with those in low-risk group. The feasibility of using the gene markers as the factors to independently predict prognosis was determined through multivariate Cox regression analysis. $P < 0.05$ indicated statistical significance. The R version 3.6.0 was adopted for all statistical analyses.

# RESULTS

## Identifying gene sets related to OS

The flow diagram of our study is summarized in Fig. 1. Univariate regression analysis was carried out on samples from the TCGA training set for the association of gene expression with patient OS. 832 genes with the log rank $p$-value of <0.01 upon univariate Cox regression analysis were found to be the underlying prognostic genes. Table 2 shows the coefficient, HR, $p$-value, and $z$-score of the 20 genes showing the highest association with OS.

## Identifying genomic variant genes harboring mutations and CNVS

Figure 2A shows gene fragments with significant amplification within the AML genome, whereas Table S1 displays genes with significant amplification within every fragment. For example, KMT2A on chromosome 11q23.3 showed significant amplification ($q$-value = 6.38E−12) (*Sakhdari et al., 2019*), ERG on chromosome 21q22.2 showed evident amplification ($q$-value = 9.50E−05) (*Canzonetta et al., 2012*), and PRSS1 on chromosome 7q34 showed marked amplification ($q$-value = 0.037333). There were 23 genes amplified in total. Figure 2B shows gene fragments with significant deletion within AML genome and Table S2 presents genes with marked deletion on every fragment. For example, ADRB2 on chromosome 5q23.1 displayed marked deletion ($q$-value = 8.54E−12), CDKN1B on chromosome 12p13.2 presented evident deletion ($q$-value = 7.28E−05). 774 genes were deleted.

Mutsig2 was used to discover genes with significant mutations at the threshold of $P < 0.05$ for mutation annotation data from TCGA. 189 significantly mutated genes were obtained. Figure 3 shows the distribution of missense mutations, synonymous mutations, framework deletion or insertion, framework displacement, splice site nonsense mutations, and additional non-synonymous mutations in the top 55 genes with the greatest significance ($P < 0.01$) within AML samples collected from TCGA. The histogram on the top stands for all non-synonymous and synonymous mutations within the 55 genes of every case, whereas the histogram on the right stands for the mutant sample number within those 55 genes. Some of the 189 genes were previously identified in prior research and are closely correlated

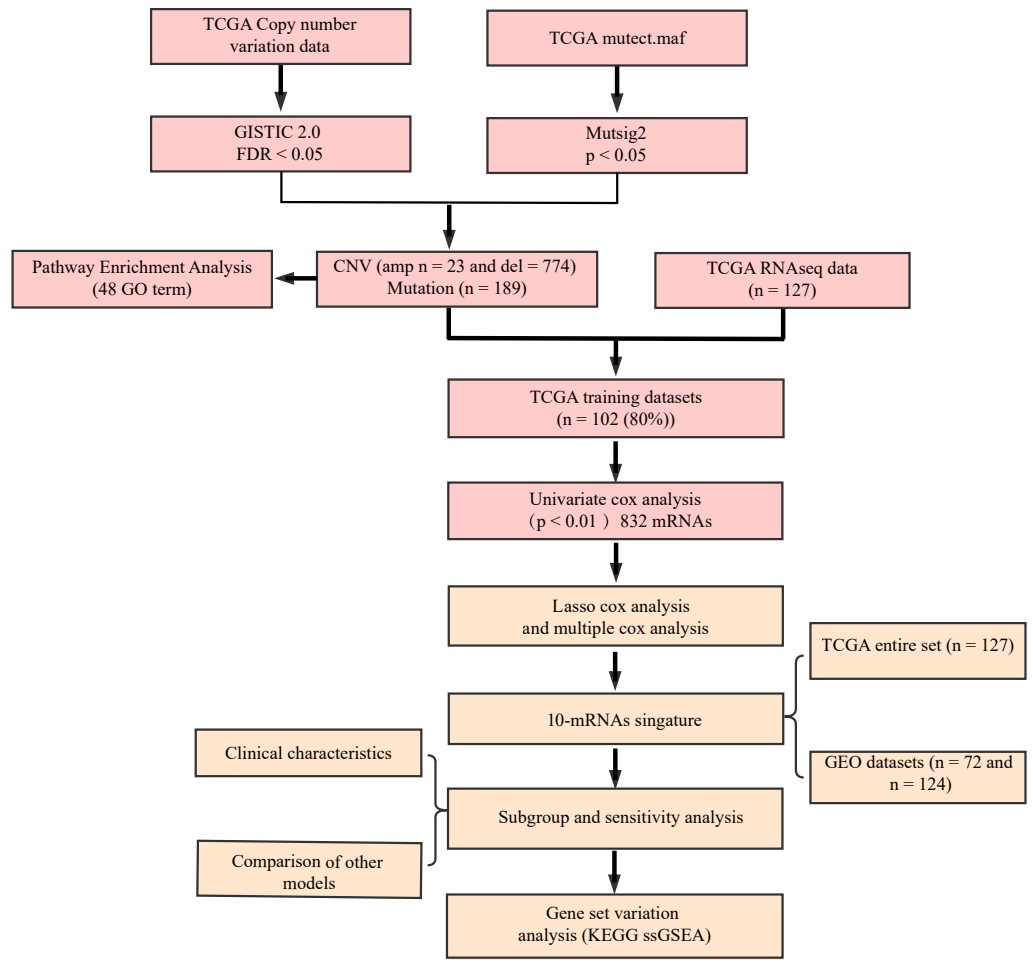

**Figure 1 Flow diagram of methods for developing the prognostic 10-gene signature.**

with tumor occurrence and progression (DNMT3A (*Yuan et al., 2019*), IDH2 (*Largeaud et al., 2019*), TP53 (*Hunter & Sallman, 2019*) and IDH1 (*Dunlap et al., 2019*)).

## Functional analyses on genomic variant genes harboring mutations and CNVs

Genes with CNV amplification and deletion and significant mutations were merged for KEGG and GO analyses to determine the functions of those genomic variant genes ($n = 977$). A total of 977 genes were enriched in AML genesis and development-related biological processes, including the myeloid leukocyte differentiation, the STAT signaling pathway, and cell–cell adhesion (Fig. 4A). 977 genes were markedly enriched in 2 pathways related to cell metabolism, including in taurine, hypotaurine, and glutathione metabolism (Fig. 4B).

**Table 2  The top 20 genes showing the greatest association with OS.**

| Gene | p value | HR | Low 95% CI | High 95% CI |
|------|---------|-----|------------|-------------|
| CBR1 | 3.85E−09 | 1.235 | 1.152 | 1.326 |
| BATF | 5.17E−07 | 1.222 | 1.13 | 1.322 |
| FIBP | 1.89E−06 | 1.154 | 1.088 | 1.224 |
| BCKDK | 2.28E−06 | 1.153 | 1.087 | 1.223 |
| MFNG | 5.05E−06 | 1.045 | 1.026 | 1.065 |
| FAM207A | 6.56E−06 | 1.206 | 1.111 | 1.308 |
| PPP2R4 | 7.72E−06 | 1.146 | 1.08 | 1.217 |
| PTP4A3 | 8.56E−06 | 1.043 | 1.024 | 1.063 |
| DNTTIP1 | 8.57E−06 | 1.192 | 1.103 | 1.288 |
| NABP2 | 9.24E−06 | 1.284 | 1.15 | 1.434 |
| PARVB | 1.09E−05 | 1.137 | 1.074 | 1.204 |
| ZNF511 | 1.48E−05 | 1.491 | 1.244 | 1.786 |
| UBE2Q1 | 1.59E−05 | 1.239 | 1.124 | 1.365 |
| TREML2 | 2.09E−05 | 1.145 | 1.076 | 1.218 |
| FERMT3 | 2.16E−05 | 1.017 | 1.009 | 1.024 |
| TGFB1 | 3.32E−05 | 1.009 | 1.005 | 1.013 |
| TNNT3 | 3.55E−05 | 1.087 | 1.045 | 1.131 |
| C7orf50 | 3.55E−05 | 1.39 | 1.189 | 1.625 |
| ATP13A2 | 4.08E−05 | 1.086 | 1.044 | 1.13 |
| DUSP7 | 4.23E−05 | 1.133 | 1.067 | 1.203 |

## Constructing the gene signature related to AML prognosis

The gene sets related to prognosis and those harboring genomic mutations and CNVs were merged and their intersection set became the potential gene set ($n = 37$). Many of these genes were not useful for clinical detection and the gene scope was further restricted to ensure greater accuracy. The glmnet function of R package was used for lasso cox regression to refine the above-mentioned prognostic genes and the gene number was reduced from 37 to 20 (Fig. 5). The MASS function of R package was used for stepwise regression analysis in accordance with the Akaike information criterion and 10 genes (Table 3) were acquired to construct a risk model.

The training set samples were divided into high and low expression groups according to the medium expression of the 10 genes and the KM curves were plotted. Significant differences in OS were observed between the high and low expression samples of 8 genes ($P < 0.05$) (Fig. S1). Among them, the low expression of FAT2, GDF9, TCERG1 and DOC2B genes was related to poor prognosis, while the high expression of CAMK2A, PREX1, DNTTIP1, C22orf42 and CRISPLD1 correlated with a terminal prognosis.

The risk score was calculated for each sample in the training set. Figure S2 shows the value of the as-constructed 10-gene signature in classifying samples from the TCGA training set. There were 45 cases in the low-risk group and 57 in the high-risk group, with a statistically significant difference (Fig. S2C) (log-rank $P < 0.0001$, HR $= 8.79$ (4.34–11.78)). The ROC curves are shown in Fig. S2B, and the AUC values at 1, 3, and 5 years are 0.92, 0.91, and 0.93,

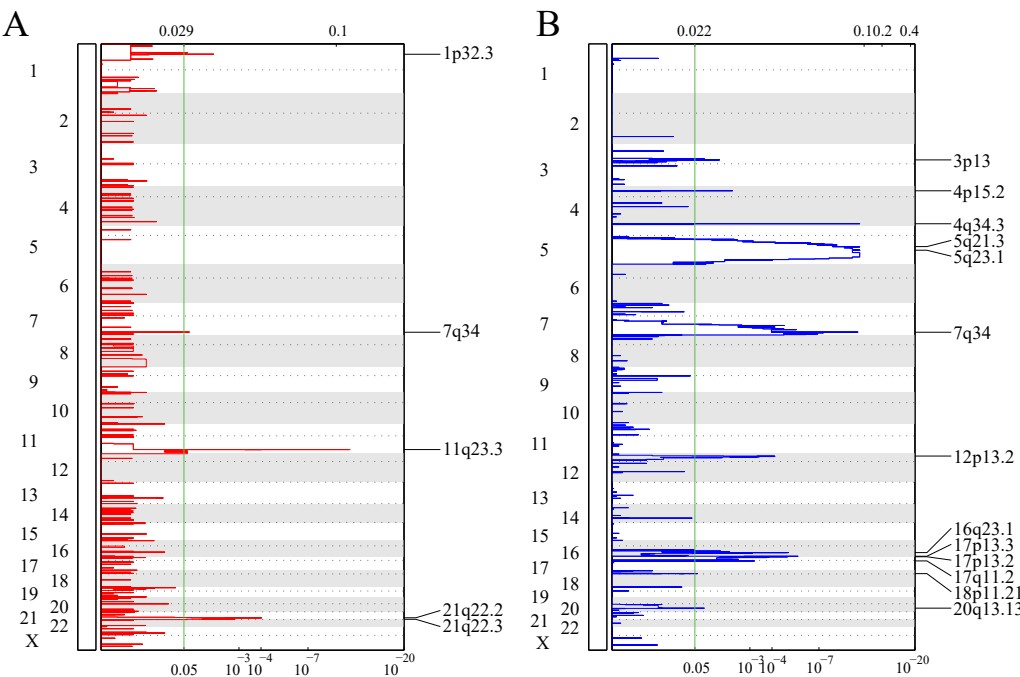

**Figure 2** **Identifying the genomic variant genes that possess CNV.** mRNAs located within the focal CNA peaks are related to AML. The false discovery rates ($q$-values) and scores obtained based on GISTIC 2.0 for alterations ($x$-axis) are plotted against the genome positions ($y$-axis); and the dotted lines stand for centromeres. (A) Gene amplifications (red) are indicated. (B) Gene deletions (blue) are displayed. Typically, the green line stands for $q$-value threshold for determining significance.

respectively. As the risk score increased, the survival time decreased and a higher instance of death was observed in the high-risk group (Fig. S2A). The high expression levels of PREX1, CAMK2A, C22orf42, DNTTIP1, CRISPLD1 and PTGIS were risk factors according to increases in the risk scores of the 10 distinct signature genes. However, high expression levels of GDF9, FAT2, DOC2B, and TCERG1 were protective and were associated with a low risk.

## Verifying the robustness of the as-constructed 10-gene signatur model

The model and threshold from the TCGA training set was used to verify the TCGA test set and to determine the robustness of the as-constructed 10-gene signature. The value in classifying samples in TCGA test set is presented in Fig. S3. There were 57 cases in the low-risk group and 70 in the high-risk group, with a statistically significant difference (Fig. S3C) (log-rank $P < 0.0001$, HR $= 6.39$ (3.62–11.29)). The ROC curves are displayed in Fig. S3B and the area value under ROC curve at 1, 3 and 5 years are 0.90, 0.87 and 0.94, respectively. Figure S3C shows similar findings as those obtained from the TCGA training set. An increase in the risk score value correlated with a decrease in the survival time to death.

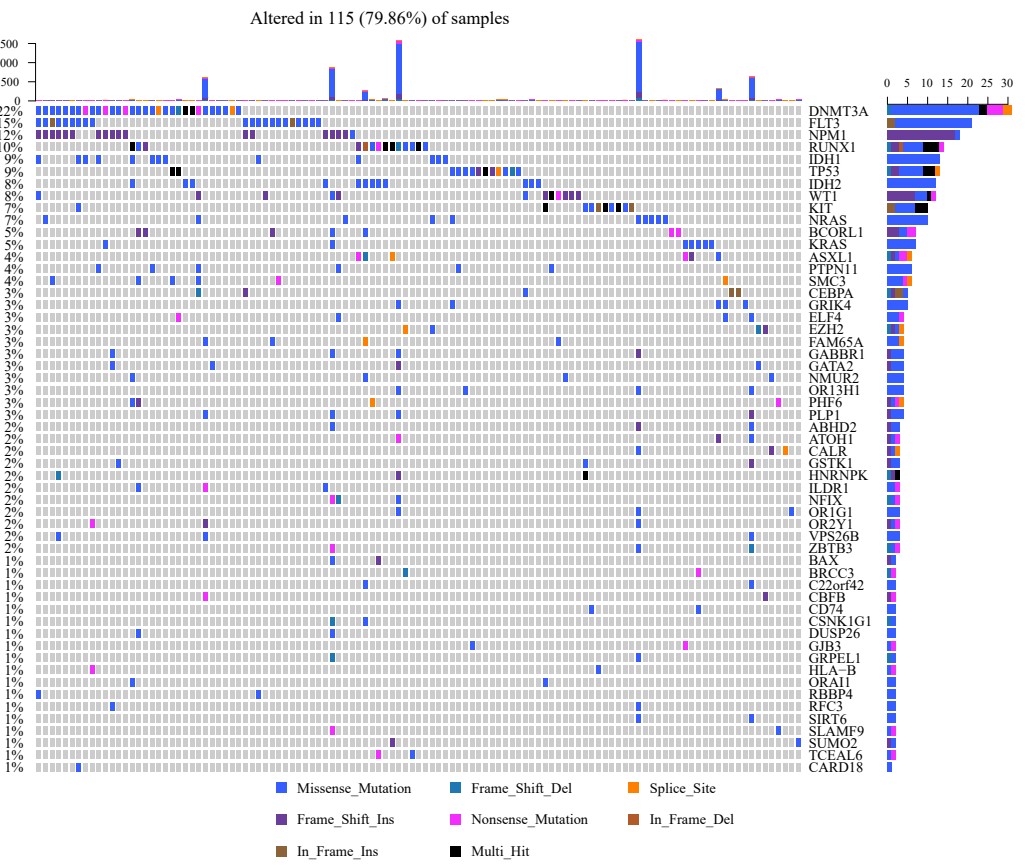

**Figure 3** **Distributions of various mutation types among those 55 genes that have the greatest significance among AML samples from TCGA.** The histogram in the top indicates the sum of non-synonymous and synonymous mutations among the 55 genes in every case, whereas the histogram on the right stands for the sample number suffering from mutantions among those 55 genes. In the heat map, the various colors stand for various mutation types, whereas the gray color represents no mutation.

Data from the GEO platform (GSE12417 and GSE37642) were selected for two external data sets to verify the value of the as-constructed 10-gene signature model in classifying data collected from diverse data platforms. Our model was used to calculate the risk score value for every sample. The threshold of 0 was used to classify samples into low- and high-risk groups after the $z$-score processing of the risk scores. A better prognosis was achieved in the low-risk group compared with the high-risk group (log-rank $p = 0.082$, HR = 1.46 (0.95–2.23); log-rank $p = 0.0051$, HR = 2.47 (1.28–4.74)) (Figs. S4C and S5C). The ROC curve analysis in Figs. S4B and S5B showed that the AUC values at 1–3 years were 0.7–0.67 and 0.72–0.62, respectively. The associations of the 10-gene expression levels with risk score were the same as those obtained in the test set and training set (Figs. S4A and S5A). Our 10-gene signature model was able to effectively predict prognosis in both the external and internal data sets.

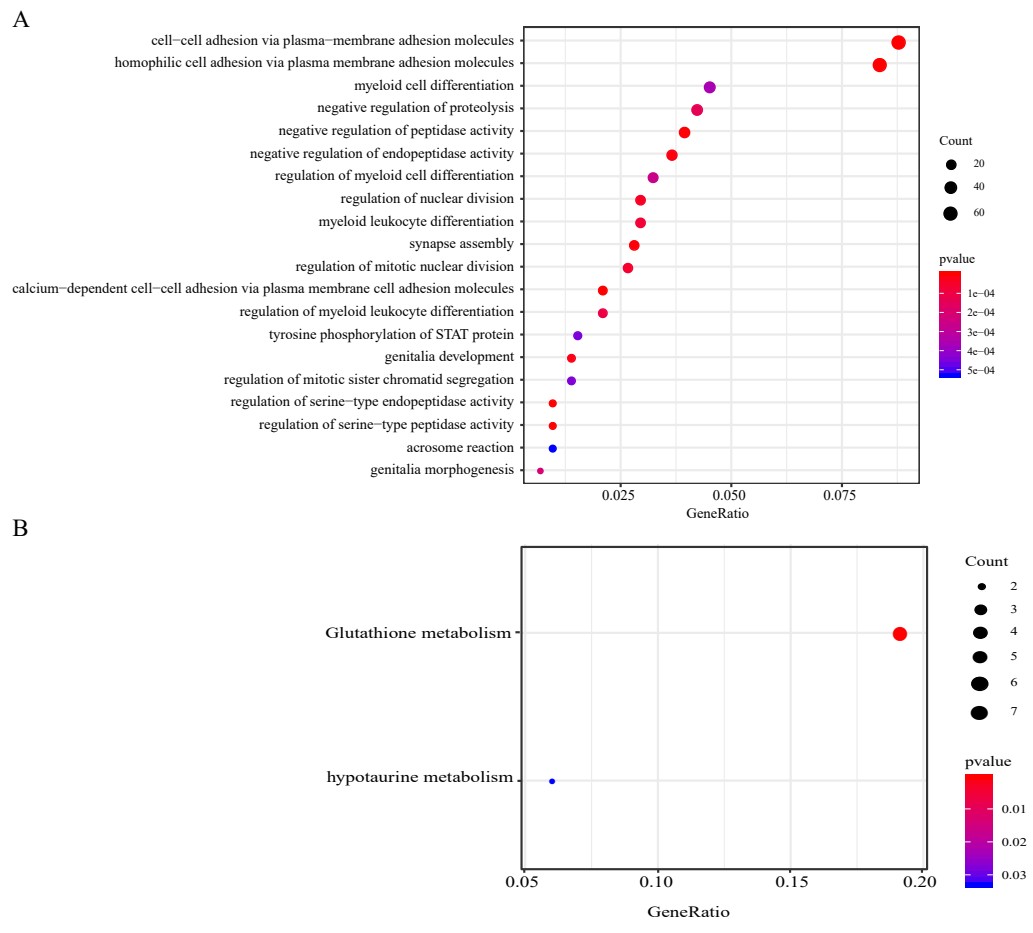

**Figure 4 Functional enrichment analyses on the 977 genomic variant genes.** (A) The GO terms that are enriched in BP category. (B) The KEGG pathways that are enriched. The diverse colors stand for distinct significance, while the various sizes indicate diverse gene numbers.

## Correlations of the 10-gene signature with sample immune and stromal scores and clinical features

A number of recent studies have reported that AML genesis and development is closely related to immune cells, stromal cell infiltration, and the microenvironment (*Boddu et al., 2018*; *Ost et al., 1985*; *Panoskaltsis, 2005*; *Karjalainen et al., 2017*). We evaluated the relationships between the 10-gene signature and the immune and stromal scores of the sample. The gene expression data was used to calculate the immune and stromal scores of each sample. Further analysis suggested that the ImmuneScore, StromalScore, and ESTIMATEScore showed significant differences between the high- and low-risk samples (classified by the 10-gene signature) in the TCGA training set (Fig. S6).

We observed the correlations of the 10-gene signature with the sample clinical features and discovered through survival analysis that the OS for TCGA training set samples was only significantly related to patient age (Fig. 6), but was not correlated with gender, bone marrow blast percentage, white blood cell count, and FAB. We also discovered that the

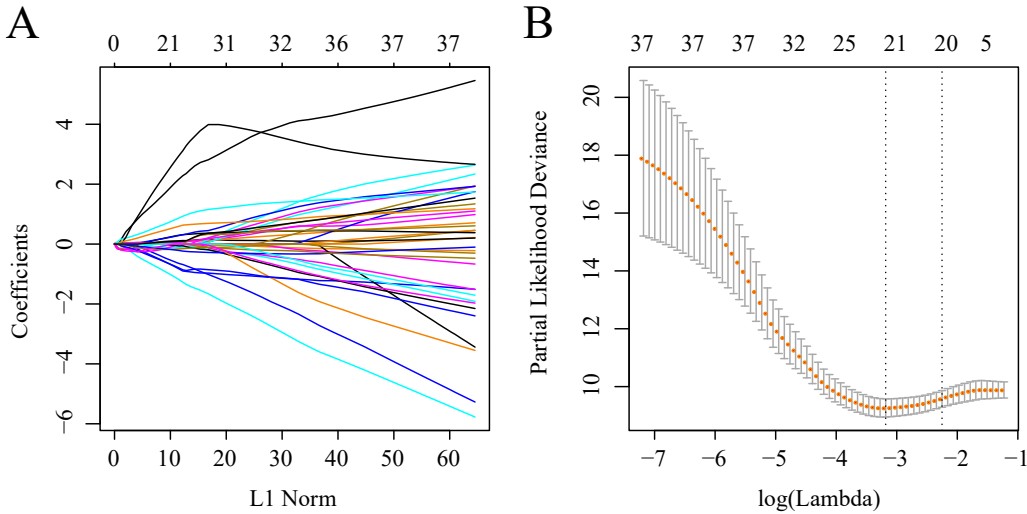

**Figure 5** **Establishment of gene signature related to AML prognosis by LASSO.** (A) The changing trajectory of each independent variable. The horizontal axis represents the log value of the independent variable lambda, and the vertical axis represents the coefficient of the independent variable. (B) Confidence intervals for each lambda.

**Table 3** **10 genes showing significant correlation with the OS of patients from training set.**

| Symbol | Coef | HR | Z-score | P value | Low 95% CI | High 95% CI |
|--------|------|-----|---------|---------|------------|-------------|
| CAMK2A | 3.1175 | 22.591 | 1.844 | 0.065 | 0.822 | 621.14 |
| FAT2 | −1.3247 | 0.265 | −2.631 | 0.009 | 0.099 | 0.713 |
| GDF9 | −1.7172 | 0.179 | −2.095 | 0.036 | 0.036 | 0.895 |
| TCERG1 | −0.907 | 0.403 | −2.019 | 0.043 | 0.167 | 0.974 |
| DOC2B | −0.4339 | 0.648 | −1.807 | 71 | 0.405 | 1.037 |
| PTGIS | 0.4833 | 1.621 | 1.953 | 0.051 | 0.998 | 2.633 |
| PREX1 | 0.426 | 1.531 | 1.884 | 0.06 | 0.983 | 2.385 |
| DNTTIP1 | 1.4097 | 4.094 | 3.204 | 0.001 | 1.729 | 9.7 |
| C22orf42 | 4.6867 | 108.492 | 3.115 | 0.002 | 5.686 | 2070.118 |
| CRISPLD1 | 0.8599 | 2.362 | 4.533 | 5.80E−06 | 1.629 | 3.427 |

10-gene signature was able to distinguish patients of different ages (young and elderly groups), different M stages (M0–M3, M4-M7), different sexes (male and female), and different white blood cell counts (WBC > 20 and WBC ≤ 20 per mcl of blood) into high- and low-risk groups. The OS revealed a significant difference between the two groups of samples ($P < 0.01$). These findings further demonstrated that our 10-gene signature displayed favorable grouping prediction capacity for patient prognosis among different clinical features (Fig. 7).

Univariate and multivariate Cox regression analyses were used to examine the clinical data from the TCGA test set, TCGA training set, and GSE37642 and GSE12417 data sets. The $p$-values, HR, and corresponding 95% CIs were analyzed to determine the capacity

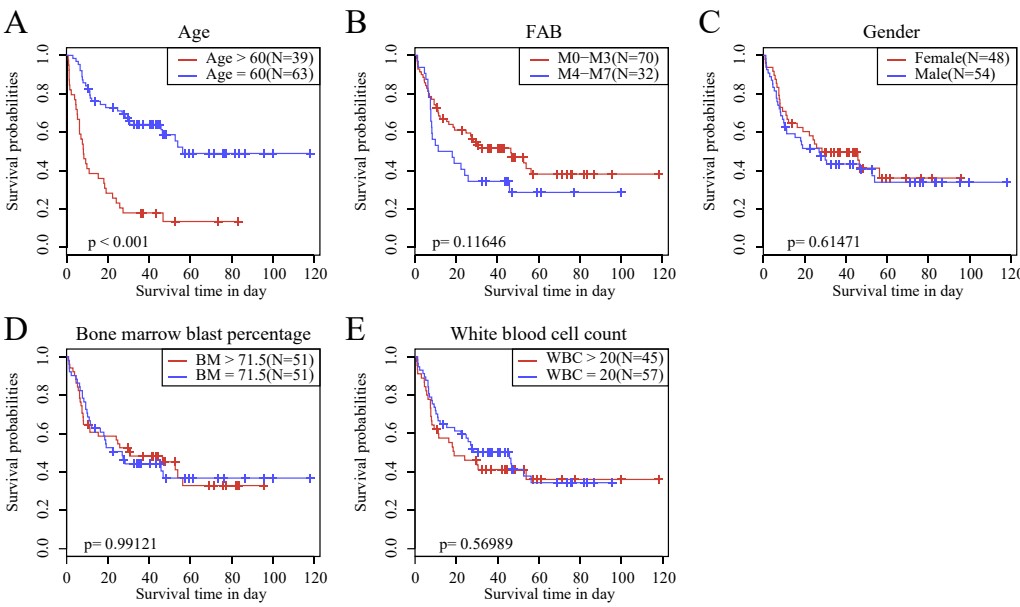

**Figure 6** **The KM curves of samples with different clinical characteristics.** (A) Age, (B) FAB, (C) gender, (D) bone marrow blast percentage, (E) white blood cell count.

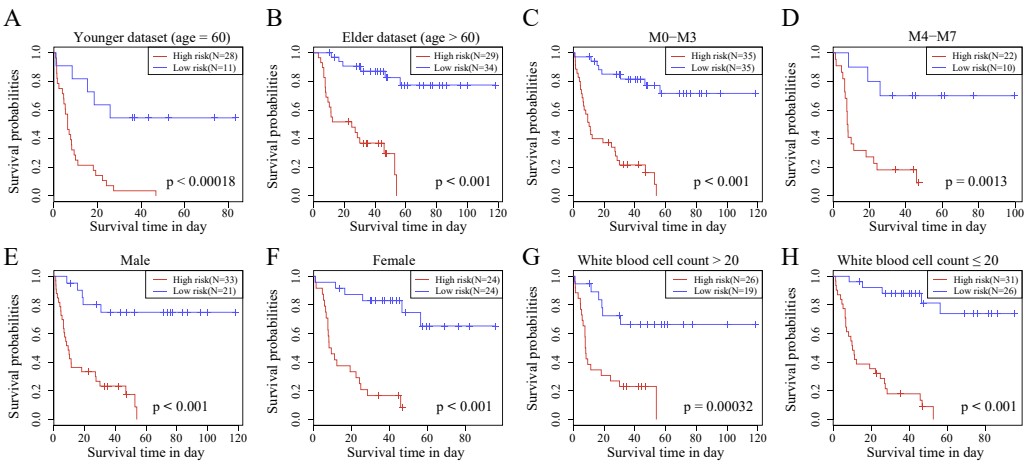

**Figure 7** **The risk distinguishability of 10-gene signature with different ages.** (A and B), M stages (C and D), sexes (E and F) and white blood cell count (G and H).

of our 10-gene signature model in independent clinical application. Clinical data from TCGA, GSE37642 and GSE12417 samples, including gender, age, white blood cell count, and FAB, were analyzed. Table 4 displays the grouping information of the as-constructed 10-gene signature. Univariate Cox regression analysis was applied to the TCGA data set. The results indicated that risk score, age, WBC, and FAB were closely correlated with survival. Multivariate Cox regression analysis revealed that age, WBC count, and risk score (HR = 5.356, 95% CI [2.970–9.658], $p = 2.42E-08$) were clinically independent. Univariate

**Table 4  Identification of clinical parameters related to patient prognosis.**

| Variables | Univariate analysis | | | Multivariable analysis | | |
|---|---|---|---|---|---|---|
| | HR | 95% CI of HR | *P* value | HR | 95% CI of HR | *P* value |
| **Entire TCGA cohort** | | | | | | |
| Risk score (High/Low) | 6.395 | 3.621–11.29 | 1.59E−10 | 5.356 | 2.970–9.658 | 2.42E−08 |
| Age | 1.036 | 1.019–1.053 | 1.61E−05 | 1.025 | 1.007–1.042 | 0.005 |
| Gender (Male/Female) | 1.001 | 0.637–1.574 | 0.994 | 0.751 | 0.462–1.219 | 0.246 |
| FAB (M0–M3 vs M4–M7) | 1.658 | 1.041–2.644 | 0.034 | 0.986 | 0.595–1.635 | 0.957 |
| Bone marrow | 0.997 | 0.986–1.009 | 0.679 | 1.004 | 0.992–1.016 | 0.512 |
| White bloodcell count | 1.005 | 0.999–1.011 | 0.05 | 1.006 | 1.001–1.012 | 0.03 |
| **GSE12417 cohort** | | | | | | |
| Risk score (High/Low) | 2.467 | 1.284–4.737 | 6.70E−03 | 2.458 | 1.273–4.746 | 7.40E−03 |
| Age | 1.031 | 1.004–1.059 | 0.022 | 1.03 | 1.004–1.056 | 0.022 |
| FAB (M0–M3 vs M4–M7) | 0.733 | 0.337–1.593 | 0.432 | 0.679 | 0.310–1.488 | 0.334 |
| **GSE37642 cohort** | | | | | | |
| Risk score (High/Low) | 1.457 | 0.951–2.232 | 8.40E−02 | 1.443 | 0.937–2.223 | 9.60E−02 |
| Age | 1.021 | 1.005–1.037 | 0.0073 | 1.02 | 1.005–1.035 | 0.01 |
| FAB (M0–M3 vs M4–M7) | 0.928 | 0.575–1.499 | 0.762 | 0.904 | 0.559–1.463 | 0.681 |

Cox regression analysis in the GSE12417 and GSE37642 data set demonstrated that age and risk score displayed a marked association with survival, whereas the multivariate Cox regression analysis revealed that age and risk score (HR = 2.458, 95% CI [1.273–4.746], $p = 7.40E−03$; HR = 1.443, 95% CI [0.937–2.223], $p = 9.60E−02$) exhibited clinical independence. The as-constructed 10-gene signature model served as an independent factor to predict prognosis and patient outcomes in a clinical setting.

## Potential signaling pathways related to the 10-gene signature

The gene expression profiles of these samples were selected for single sample GSEA using the GSVA function of R package to further reveal the signaling pathways involved in regulating AML patient prognosis and to observe the relationships between the sample risk score and the regulatory signaling pathways (*Hanzelmann, Castelo & Guinney, 2013*). Most signaling pathways were negatively correlated with sample risk score but a small fraction were positively correlated with the risk score (Fig. 8A). The 24 KEGG pathways with the correlation significance of >0.35 were selected and clustering analysis was performed according to their enrichment scores (Fig. 8B). Among the 24 pathways, cell adhesion molecules cams and hematopoietic cell lineage increased with higher risk scores, whereas the Hedgehog signaling pathway decreased with the higher risk score. These results show that the dysregulation of these pathways was closely related to the development and progression of AML.

## Comparisons between the 10-gene signature risk model with other AML prognosis prediction model reported in literature

We selected 4 prognosis-related risk models based on a review of the literature. These included models with an 11-gene signature (*Huang, Liao & Li, 2017*), a 6-gene
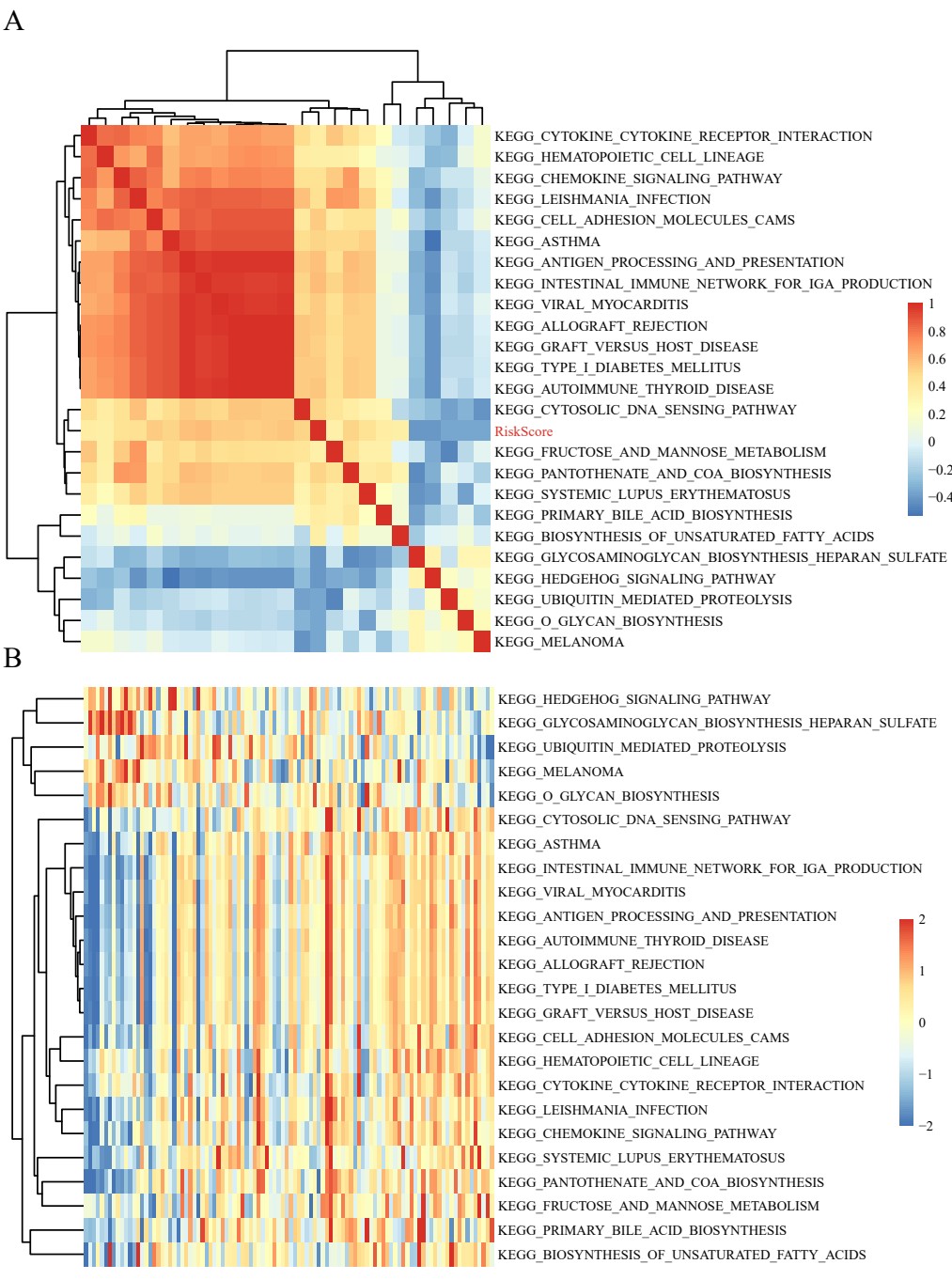

**Figure 8   The correlations of enriched signaling pathways with risk score.**

signature (*Zhao, Li & Wu, 2018*), a 6-gene signature (*Zhang et al., 2019*) and a 4-gene signature (*Nguyen et al., 2019*) to compare with our 10-gene signature model. The risk score of each AML sample in the TCGA database was calculated according to the same method based on the levels of corresponding genes in these 4 models to make the model

comparable. The samples were divided into Risk-H and Risk-L groups according to the medium risk score to calculate the OS difference between two groups of samples. The ROC and OS-KM curves of the 4 models are displayed in Fig. 9. It was observed that the 1–5 year AUC values of the 4 gene models were above 0.71 but that their predictive accuracy was slightly inferior to that of our 10-gene signature. We used the rms package to calculate the concordance index (C-index) values of these 5 models to further compare the prediction performance of these models on the AML samples. Our 10-gene signature had the highest C-index (Fig. 10A) among the 5 models, suggesting that it had superior overall prediction performance. The restricted mean survival (RMS) can be explained as the mean free event survival time within a specific time period, which is equal to the area under the KM curve at a specific time point. The RMS time was used to evaluate the prediction performances of the 5 models at different time points. The RMS curves showed that the 10-gene risk model was superior to the models proposed by Li, Wu, Deng, and Heller (Fig. 10B) for a period >80 months, as suggested by the AUC values of these 5 models.

## DISCUSSION

AML can be diagnosed and treated in its early stages. However, the conventional clinicopathological factors including stage, age, and WBC counts cannot be used successfully in a predictive fashion (*Dohner, Weisdorf & Bloomfield, 2015*). There is no effective universal therapeutic strategy for stratifying risk. Therefore, it is important to screen the prognostic molecular markers that comprehensively reflect the biological characteristics of cancer for a more individualized approach to patient care for AML. Our study examined the expression patterns in AML samples from the GEO and TCGA databases. We constructed a reliable OS-related 10-gene signature that was not dependent on clinical parameters.

Several gene signatures are currently used in clinical practice including the Oncotype DX for grading the risk of disease relapse on the basis of 21 gene expression levels in breast cancer (*Chen et al., 2013*) and the Coloprint that is established according to 18 gene expression levels in colon cancer (*Tan & Tan, 2011*). The above findings indicate that new cancer prognostic markers identified according to gene expression data is a prospective high-throughput identification approach at the molecular level. Systemic biological approaches are used to identify the genetic biomarkers related to AML prognosis and to construct the genetic features. However, the AUC value at 3–5 years can be low when using an external data set or an excessive gene number is occasionally observed, which may disadvantage big data promotion and verification. Our as-constructed 10-gene signature achieved a high AUC value with a low gene number and can be applied clinically. PREX1, CAMK2A, C22orf42, DNTTIP1, CRISPLD1 and PTGIS served as risk factors and GDF9, FAT2, DOC2B and TCERG1 were protective factors. 9 out of 10 (PREX1, CAMK2A, DNTTIP1, CRISPLD1, PTGIS, GDF9, FAT2, DOC2B and TCERG1) genes are typically suggested as biomarkers in several cancers and show close correlations with multiple cancer prognoses, including in endometrial cancer, breast cancer (BC), ovarian cancer, lung cancer, oral cancer, and colorectal cancer (CRC)
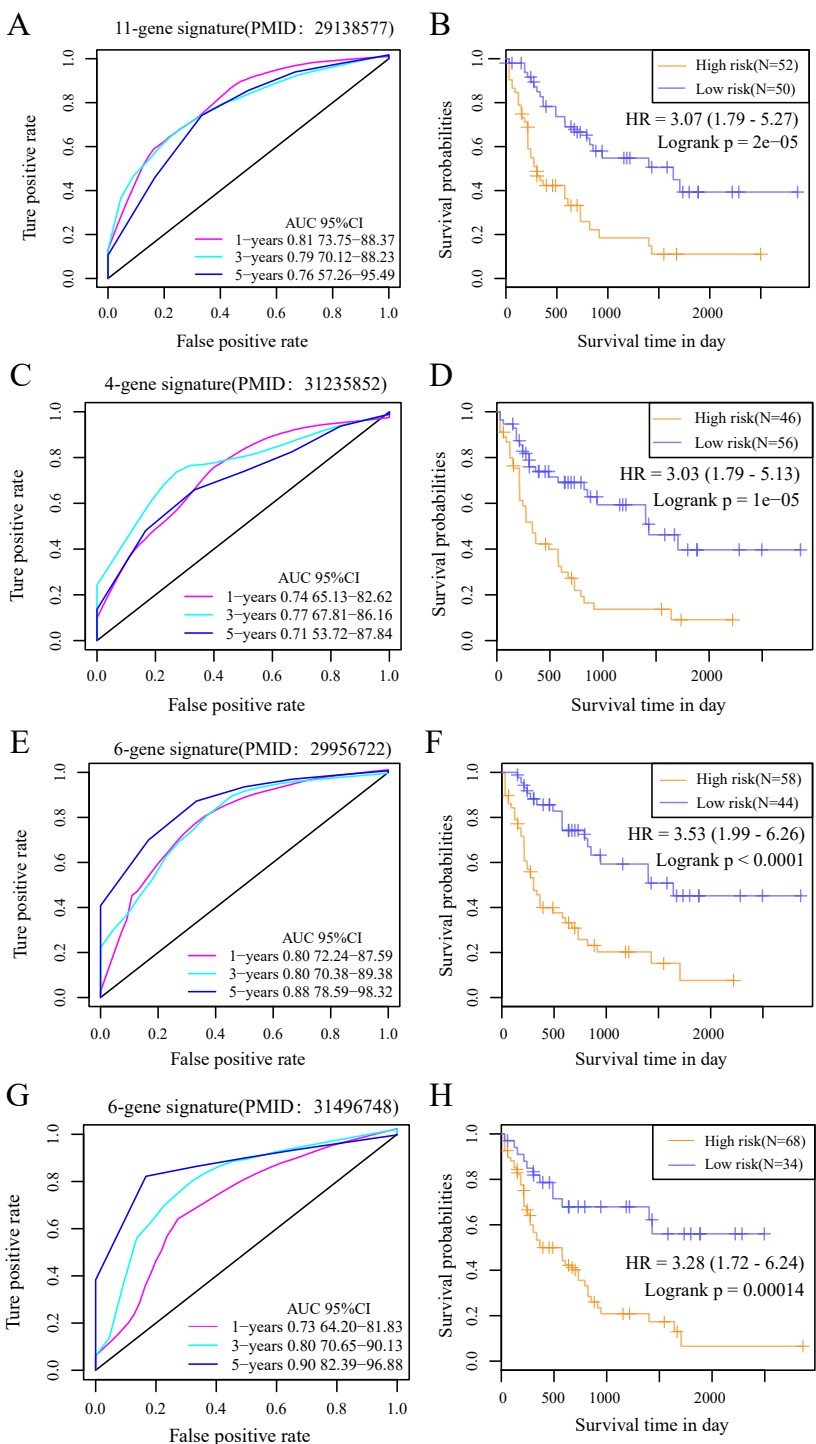

**Figure 9** **The AUC and prognosis KM curve of 11-gene signature.** (PMID: 29138577, A), 4-gene signature (B, PMID: 31235852), 6-gene signature (C, PMID: 29956722) and 6-gene signature (D, PMID: 31496748) in predicting the Risk-H and Risk-L groups on the training set.

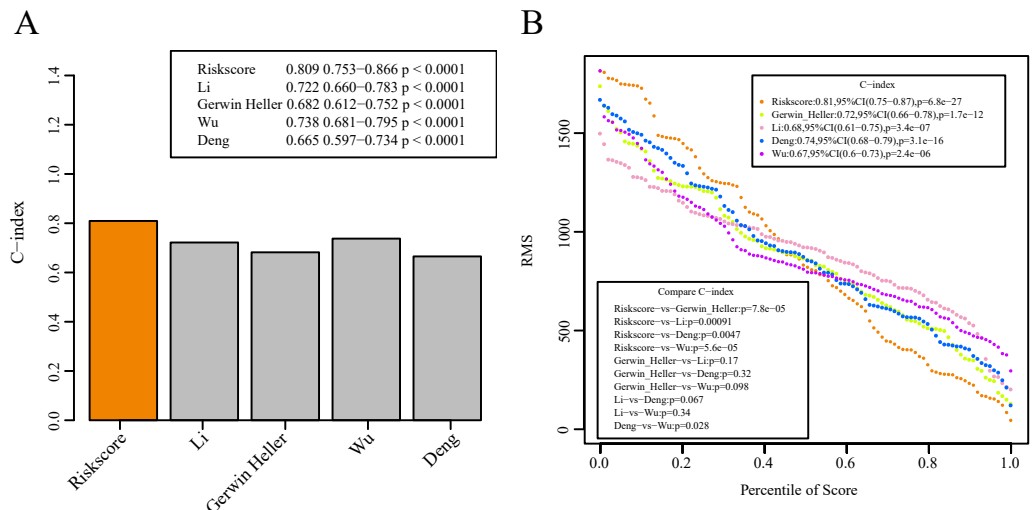

**Figure 10 The prediction performance of these five models on the AML samples.** (A) The concordance index (C-index) values of these 5 models. (B) The RMS curves of these five models.

(*Dillon & Miller, 2015*; *Chen et al., 2017*; *Sawai et al., 2018*; *Li et al., 2019*; *Singh, Chandra & Bapat, 2015*; *Du et al., 2012*; *Katoh, 2012*; *Satyamoorthy et al., 2014*).

The identified genes are closely correlated to tumor prognosis. Our study was the first to determine these new prognostic markers for AML. The results of GSEA suggest that the 10-gene signature-enriched pathways were closely associated with the biological processes and pathways involved in the development of AML (*Savona et al., 2018*). Such findings indicate that the as-constructed model may be a useful clinical tool and may be a potential diagnostic target for patients.

The bioinformatic technique was used to identify the possible prognosis-related genes within a large number of samples. Nonetheless, there were limitations to our study. Some samples did not have complete clinical follow-up data, so it was impossible to examine the feasibility of using these biomarkers in distinguishing patient prognosis according to additional health factors. The bioinformatic analysis results alone were insufficient and additional large-scale and genetic study experiments must be performed to verify the results.

A prognosis stratification system based on the 10-gene signature was proposed in our study and showed a favorable AUC in the verification and training sets. It was independent of clinical features. The gene classifier is relative to clinical features and contributes to improving the prediction accuracy of survival risk and is recommended as a molecular diagnostic test for evaluating AML prognosis.

### Funding

This work was supported by the Natural Science Foundation of Zhejiang Province (LY19H290003, LQ20H280002); the Zhejiang Provincial Medical and Health Science and Technology Project (2020KY196, 2018277310); the Foundation of Zhejiang province Chinese medicine science and technology planes (2017ZB030, 2020ZA044); and Key project of the 2017 school research fund of Zhejiang Chinese Medical University (2017ZZ02). The funders had no role in study design, data collection and analysis, decision to publish, or preparation of the manuscript.

### Grant Disclosures

The following grant information was disclosed by the authors:
Natural Science Foundation of Zhejiang Province: LY19H290003, LQ20H280002.
Zhejiang Provincial Medical and Health Science and Technology Project: 2020KY196, 2018277310.
Foundation of Zhejiang province Chinese medicine science and technology planes: 2017ZB030, 2020ZA044.
Key project of the 2017 school research fund of Zhejiang Chinese Medical University: 2017ZZ02.

### Competing Interests

The authors declare there are no competing interests.

### Author Contributions

- Haifeng Zhuang conceived and designed the experiments, analyzed the data, authored or reviewed drafts of the paper, and approved the final draft.
- Yu Chen and Xianfu Sheng conceived and designed the experiments, performed the experiments, analyzed the data, authored or reviewed drafts of the paper, and approved the final draft.
- Lili Hong, Ruilan Gao and Xiaofen Zhuang performed the experiments, analyzed the data, prepared figures and/or tables, and approved the final draft.

### Data Availability

All the data we used in our study are available from NCBI GEO: GSE37642, GSE12417 and The Cancer Genome Atlas (TCGA).

The UCSC Cancer Browser was utilized to download CNV, clinical follow-up, TCGA RNA-Seq and SNP 6.0 chip data from TCGA: https://xenabrowser.net/datapages/?cohort=GDC%20TCGA%20Acute%20Myeloid%20Leukemia%20(LAML)&removeHub=http%3A%2F%2F127.0.0.1%3A7222.

### Supplemental Information

Supplemental information for this article can be found online at http://dx.doi.org/10.7717/peerj.9437#supplemental-information.

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
