# Peer review of "Searching for a signature involving 10 genes to predict the survival of patients with acute myelocytic leukemia through a combined multi-omics analysis"

_PeerJ, doi:10.7717/peerj.9437_

## Round 0.1 · original submission · Major Revisions

The authors needs to answer each of the reviewers comments. Especially the methods they applied need clarification by addressing each of the concerns raised by the reviewers. Finally, the manuscript should be checked and redacted by a native speaker.

Reviewer 1 ·

Basic reporting

Complete review submitted under "General comments to the author."

Experimental design

Complete review submitted under "General comments to the author."

Validity of the findings

Complete review submitted under "General comments to the author."

Additional comments

By employing data garnered from public databases, the group describe establishment of a predictive 10-gene signature profile for over-all survival in acute myeloid leukaemia.
The findings are consistent with previously published data, with minor progress in understanding within the field.

Major points
Several major comments which require either supporting data and/or written clarification before acceptance for publication are:
1/ A fundamentally important criteria is the establishment of cut-off values in categorising low v high expression and hence low v high risk.
Line 189: “The training set samples were divided into high and low expression groups according to the medium expression of these 10 genes, ...”
This is unclear.
The authors need to explicitly justify the cut-off criteria and explain the imbalance of the dataset between the training versus test sets – high-v-low risk Fig5 57v45, Fig6 70v57, Fig7 37v35 and Fig8 66v58 (which may have introduced bias within the analyses.)
Other arbitrary cut-offs are applied with WBC (set at 20) and staging (M0-M3 v M4-M7)

2/ Similarly, the authors fail to rationalise why comparison to the other models is made at 80-months and beyond (Figure 14). Given the patient and disease evolution biology, it is unlikely that diagnostic material from 6 years previously will be used in patient management strategies.
3/ Although data may be limited, substantial knowledge may be gained if the gene signature was determined as a measure of relapse/resistance to therapy.


Minor points
Given the central point of this report, the construction of the 10-gene signature requires further elaboration, with more detail of the steps employed in the R-package as well as analysis applied in accordance to the AIC.
Although the reporting is on the whole clear and systematic, with supporting explanations; the manuscript requires re-writing by a native English speaker with scientific knowledge in the field. Apart from the abstract, several phrases and reference points are inconsistent and ambiguous e.g. Lines 61-62, 214, 216, 235, 285 and throughout the ‘Discussion’.

Figure labels lacking:
Figure 5-8
A: lower chart referring to overall survival: X-axis label is lacking. Is this log or linear expression value? Can the heat-map be segregated into low v high risk strata?

Although informative, all 14 figures do not need to be within the main body of the manuscript. Some of the figures (for example from Figures 5-8) could be migrated to the supplementary sections.

What are the units for WBC?

Reviewer 2 ·

Basic reporting

The English language used needs significant editing. Although the text is understandable, it contains unusual word usage, grammar errors, typos, and long, complicated sentences. Already in the title:

Title “survival for patients” => “survival of patients”
Title “through the combined” => “through a combined”
Line 19 “multi-omics datas” => “multi-omics data”

The background is properly referenced and previous similar signatures are discussed. However, the paper contains an extensive number of figures, that are unnecessary and some of most of them are unreadable. Figure 1 and 2 are summaries of already existing results from TCGA. This is unnecessary. Figure 3 is almost unreadable and the colors are hard to see. Figure 4 doesn’t add much information to the paper, Figure 5 – 8 have exactly the same layout and data type showing different datasets. These last 4 figures should be summarized into a single figure, showing the most important finding. Figure 9 should be a supplementary material figure. Figure 10 and 11 contain 13 KM curves. Again, these should be condensed into a single figure/panel showing the most relevant information, illustrating the paper’s results. Figure 12, 13, and 14 are probably the most important results, but they also have a low resolution, with hard to read text.

Experimental design

The experimental design is acceptable, even if the paper only uses already published data, it is analyzing this data in a new way. However, the methods used are unclear.

- When did the authors download the data, and exactly which version?
- What does “TCGA RNA-Seq data of SNP 6.0 chip” on Line 96 mean? Are the authors talking about RNA-seq data or SNP chip data?
- How did they process the GEO datasets? Apparently, these GEO datasets contain raw CEL files, and the authors do not describe any data processing steps for them.
- What does “those with marked correlation with patient overall survival” on Line 106 mean?
- On Line 104: how was the analysis carried out. OS was correlated with what? Mutation status? Expression level? Something else?
- The description of methods is confusing at times. For example, the authors write in Line 109 “this study used the GISTIC 2.0 software”. Did the authors use GISTIC themselves, or only refer to the methods used by TCGA?
- The above question is true for the Mutsig analysis.
- On Line 119 and 126 the authors write several times “… function of R package”. What does this mean? There is no general “R package”. The text needs to be clarified, to include the specific functions used from specific R packages.
- The functional enrichment analysis is not described in detail.
- How was high-risk and low-risk defined?
- The results section also contains some information that should go into the Methods section, for example Line 148, 196, 235, 272.
- The exact version of the various R packages is missing.
- On Line 196, what does “z-score processing” means?
- Unclear how the stromal, immune and estimate scores were calculated for Figure 2. Line 235 mentions an “estimate function of R package” but there is no such function in R.
- On Line 272, how was the GSVA score calculated and how were the signaling pathways defined?

Validity of the findings

It is hard to assess the validity of these findings, as the Methods section lacks important information and the included descriptions are unclear. The large number of Figures, without highlighting the most important details, is also confusing.

---

## Round 0.2 · accepted · Accept

The authors have addressed all of the concerns raised by the reviewer. The manuscript is now acceptable for publication.

Reviewer 1 ·

Basic reporting

The authors have sufficiently addressed all major concerns and have significantly improved the manuscript to the level fit for the journal.

Experimental design

Consistent with the standard protocols and described with the necessary detail.

Validity of the findings

Sufficiently addressed in the rebuttal.

Additional comments

Acceptable for publication.